# An Assessment of the Viability of Lytic Phages and Their Potency against Multidrug Resistant *Escherichia coli* O177 Strains under Simulated Rumen Fermentation Conditions

**DOI:** 10.3390/antibiotics10030265

**Published:** 2021-03-05

**Authors:** Peter Kotsoana Montso, Caven Mguvane Mnisi, Collins Njie Ateba, Victor Mlambo

**Affiliations:** 1Antimicrobial Resistance and Phage Biocontrol Laboratory, Department of Microbiology, Faculty of Natural and Agricultural Sciences, North-West University, Private Bag X2046, Mmabatho 2735, South Africa; Collins.Ateba@nwu.ac.za; 2Food Security and Safety Niche Area, Faculty of Natural and Agricultural Sciences, North-West University, Private Bag X2046, Mmabatho 2735, South Africa; 23257539@nwu.ac.za; 3Department of Animal Sciences, Faculty of Natural and Agricultural Sciences, North-West University, Private Bag X2046, Mmabatho 2735, South Africa; 4School of Agricultural Sciences, Faculty of Agriculture and Natural Sciences, University of Mpumalanga, Private Bag X11283, Mbombela 1200, South Africa; Victor.Mlambo@ump.ac.za

**Keywords:** *E. coli* O177, foodborne pathogen, ex vivo rumen fermentation, animal infection, bacteriophage therapy

## Abstract

Preslaughter starvation and subacute ruminal acidosis in cattle are known to promote ruminal proliferation of atypical enteropathogenic *Escherichia coli* strains, thereby increasing the risk of meat and milk contamination. Using bacteriophages (henceforth called phages) to control these strains in the rumen is a potentially novel strategy. Therefore, this study evaluated the viability of phages and their efficacy in reducing *E. coli* O177 cells in a simulated ruminal fermentation system. Fourteen phage treatments were allocated to anaerobic serum bottles containing a grass hay substrate, buffered (pH 6.6–6.8) bovine rumen fluid, and *E. coli* O177 cells. The serum bottles were then incubated at 39 °C for 48 h. Phage titres quadratically increased with incubation time. Phage-induced reduction of *E. coli* O177 cell counts reached maximum values of 61.02–62.74% and 62.35–66.92% for single phages and phage cocktails, respectively. The highest *E. coli* O177 cell count reduction occurred in samples treated with vB_EcoM_366B (62.31%), vB_EcoM_3A1 (62.74%), vB_EcoMC3 (66.67%), vB_EcoMC4 (66.92%), and vB_EcoMC6 (66.42%) phages. In conclusion, lytic phages effectively reduced *E. coli* O177 cells under artificial rumen fermentation conditions, thus could be used as a biocontrol strategy in live cattle to reduce meat and milk contamination in abattoirs and milking parlours, respectively.

## 1. Introduction

Atypical enteropathogenic *E. coli* (aEPEC) group contains heterogeneous serotypes, which causes severe illness in humans [1,2]. Recent studies have reported that some serotypes from this group harbour stx operon (*stx*_1_ and *stx*_2_) responsible for production of Shiga toxin [2,3]. Shiga-toxigenic aEPEC may cause life-threatening conditions such as haemorrhagic colitis (HC) and haemolytic uremic syndrome (HUS) in humans [4]. Although aEPEC strain is known to inhabit the lower gastrointestinal tract (GIT) of healthy cattle, preslaughter starvation and subacute ruminal acidosis have been reported to promote their proliferation in the rumen [5,6,7,8,9]. This may contribute substantially to the amount of microbial load present in the lower GIT of a live animal. High loads of shiga toxigenic multidrug resistant aEPEC in the GIT of cattle increase the risk of recurrent contamination of milk and meat (carcass) during milking and slaughter [10,11,12,13]. As a result, this may pose a substantial food safety risk.

Although a variety of biological, physical, and chemical methods have been used to reduce the level of foodborne pathogens such as *E. coli* species on food, these decontamination methods have practical drawbacks [14]. For example, irradiation leads to lipid oxidation, and consequently, changes in organoleptic properties of food [15] while the use of chlorine produces carcinogenic trihalomethanes [16]. Furthermore, application of these methods, especially in live animals prior to slaughter is cumbersome [17]. Consequently, animals with high microbial load may contribute to cross-contamination of the carcass during slaughtering process. Against this backdrop, development of novel strategies to control the proliferation of *E. coli* pathogens in live animals to prevent food contamination and subsequent human infections is urgently needed [18]. One such novel strategy is the use of phages to prevent the proliferation of shiga toxigenic multidrug resistant (MDR) aEPEC strain in live cattle.

Phages are viruses that infect and kill bacteria cells through a lytic process [19,20]. The phages are host-specific, self-replicating, and harmless to eukaryotic cells and cause minimal or no effect to nonhost microbiota [21,22,23,24]. In addition, phages can kill multidrug resistant bacteria strains, which pose significant public health risks [15]. Based on these attributes, lytic phages are regarded as a viable alternative to combat the spread of multidrug resistant *E. coli* O177 strain. In the “bio-control intervention strategy”, phages can be applied on live animals to reduce the level of foodborne pathogens, thereby improving food safety [25]. Some studies have indicated the potential of lytic phages against *E. coli* O157 in live calves [18,26]. Furthermore, studies with live animals have focused on the use of phages against shiga-toxigenic *E. coli* O157 and other non O157 serotypes [26]. Indeed, no study has assessed the potency of single phages and/or phage cocktails against shiga-toxigenic MDR *E. coli* O177 cells in the rumen. This strain was selected because it harbours virulence and antibiotic resistance genes (3). Furthermore, its presence in cattle may result into food contamination during slaughter and milking. Therefore, this study was designed to evaluate the viability and efficacy of single phages and their cocktails against *E. coli* O177 cells in an in vitro ruminal fermentation system. We tested the hypotheses that single phages, and their cocktails can survive and thus reduce *E. coli* O177 cells under simulated ruminal fermentation conditions.

## 2. Results

### 2.1. Viability of Phages and Time-Induced Changes in Total E. coli O177 Cell Counts

Both single phages and phage cocktails were viable and stable under rumen simulation conditions. Figure 1 presents the number (mean log_10_) of phages (PFU/mL) recovered from artificial rumen contents over a 48-h incubation period. The titres of single phages and phage cocktails show quadratic increases (*p* < 0.05) over incubation time, indicating viability in the simulated rumen fermentation conditions. As presented in Table 1 and Table 2, both single phage and phage cocktail titres were projected to peak after 49–52 h and 51–55 h of incubation as calculated from quadratic equations with R^2^ values ranging from 0.811 to 0.994 and 0.982 to 0.995, respectively. Phage vB_EcoM_3A1’s titre was predicted to peak after 49 h of incubation while vB_EcoM_10C3’s titre was predicted to reach a maximum after 53 h of incubation. For phage cocktails, vB_EcoMC1’s titre was predicted to peak after 51 h of incubation whereas vB_EcoMC4’s titre was projected to peak after 55 h of incubation.

### 2.2. Efficacy of Phages against E. coli O177 Cells

As shown in Table 3 and Table 4, both single phages and phage cocktails reduced *E. coli* O177 cell counts over the 48-h incubation period (*p* < 0.05). *E. coli* O177 cell count reduction ranged from 16.03% to 62.74% in samples treated with single phages. Phage vB_EcoM_366B was the most effective against *E. coli* O177 cells at 6 h (19.41%) and 12 h (28.61%) post-inoculation. At 24 h post-inoculation, phage vB_EcoM_11B2 was the most effective resulting in 50.10% reduction in *E. coli* O177 cell counts while the highest effect at 36 h (56.42%) and 48 h (62.74%) was recorded in samples treated with the phage vB_EcoM_3A1 (*p* < 0.05). The number of *E. coli* O177 cell counts ranged between 20.65% and 66.92% in samples treated with phage cocktails over the 48-h incubation period. Phage cocktail vB_EcoMC4 was the most effective phage cocktail in reducing *E. coli* O177 cell counts at 6 h and 12 h, with the maximum reduction of 26.63% and 12 h 36.75%, respectively while at 24 h and 36 h, the highest (55.19% and 60.69%, respectively) reduction was recorded in samples treated with vB_EcoMC6. At 48 h, the most effective cocktail against *E. coli* was vB_EcoMC4 that caused a 66.92% reduction in cell counts.

Relationships between time of incubation and percent reduction in *E. coli* O177 cell counts when challenged with single phages and phage cocktails are presented in Table 5 and Table 6. For single phages, efficacy against *E. coli* O177 cells peaked (60.81–63.27%) at 47–48 h of incubation as determined from prediction equations with R^2^ values ranging from 0.992 to 0.996. The highest efficacy (62.73%) after 48 h was observed in samples treated with vB_EcoM_12A1 while the lowest efficacy (61.031%) was seen in samples treated with vB_EcoM_10C2 (*p* < 0.05). The maximum efficacy (63.27%) at 47 h was seen in *E. coli* challenged with vB_EcoM_366B whereas the minimum efficacy (60.81%) was observed with vB_EcoM_10C3. After exposure to phage cocktails, efficacy peaked (63.06–73.25%) at 43–46 h of incubation as determined from prediction equations with R^2^ values ranging from 0.970 to 0.993. For phage cocktails, vB_EcoMC3 and vB_EcoMC6, efficacy against *E. coli* O177 reached a maximum (73.25% and 66.90%, respectively) at 43 h whereas for vB_EcoMC2 and vB_EcoMC4, efficacy peaked (63.80% and 67.18%, respectively) at 44 h of incubation (*p* < 0.05). For phage cocktail vB_EcoMC1, efficacy peaked (66.06%) at 46 h while for cocktail vB_EcoMC5, efficacy against of *E. coli* O177 reached a maximum (63.06%) after 45 h (*p* < 0.05).

## 3. Discussion

Most farm animals, including ruminants, harbour foodborne pathogens [3,27,28,29]. In addition, intensively reared beef and dairy animals tend to be housed in large numbers and this may promote proliferation and dissemination of foodborne pathogens into the environment [30]. Furthermore, supper-shedders may increase the risks of preharvest cross contamination of beef carcasses and milk intended for human consumption [31,32,33]. Given the limitations of traditional decontamination methods, the use of phages is considered as a practical alternative to reduce *E. coli* pathogens in live animals [34]. Several studies have evaluated the ability of phages to reduce foodborne pathogens, especially *E. coli* O157:H7 in live animals [18,26,35,36,37]. In vitro rumen fermentation models have only been used to investigate the efficacy of phages against *E. coli* O157 [36,37], but never for *E. coli* O177 strain. The current study evaluated the stability and efficacy of single phages and phage cocktails in reducing *E. coli* O177 cells under simulated ruminal fermentation conditions over a 48-h incubation period. Notably, all single phages and phage cocktails were stable under simulated ruminal fermentation (pH 6.3, 39 °C and CO_2_). Following phage treatment, *E coli* O177 cell counts significantly decreased while phage titres increased with incubation time. Bacterial cells infected by phages tend to release many virions (phage particles) [38] resulting in higher phage titres that were observed over time in this study. Contrary to the findings by Bach et al. [37] who reported a significant decrease of phage titres even after 48 h of incubation, our study showed that single phage titres increased with incubation time and were predicted to attain their maximum titre values after 49–53 h of incubation. This difference between the two studies could be attributed to different bacteria host targeted (*E. coli* O177 vs. *E. coli* O157:H7), the amount of *E. coli* added to the fermentation bottles (10^8^ CFU/mL vs. 10^4^ CFU/mL), the in vitro ruminal fermentation technique (Reading Pressure Technique vs. Rusitec), and basal substrates (*Eragrostis plana* hay vs. barley and silage) used. Phages intended for biocontrol purpose should be stable under various environmental conditions [39]. The results obtained in this study revealed that single phages required the shortest incubation time (49–53 h) to attain peak titres compared to phage cocktails (51–55 h). This could be due to the emergence of phage resistant *E. coli* O177 cells, which may hinder the phage infection process, especially in the samples treated with single phages [40]. Interestingly, both phage types were stable under simulated rumen conditions (39 °C, pH 6.5, anaerobic condition with the presence of other rumen microbes and enzymes). This suggests that these are suitable candidates for reducing *E. coli* O177 cells in cattle prior to slaughter [34]. As a result, this may not only reduce microbial load in live animals but also mitigate contamination of carcass during slaughter and thus minimise meat waste, which normally occurs during slaughter-trimming process of contaminated areas on beef carcasses.

Based on the results obtained in this study, the highest phage potency against *E. coli* O177 cell was observed between 47 h and 48 h (for single phages) and between 43 h and 46 h (for phage cocktails). This suggested that phage cocktails required a short time to reduce bacterial cell counts as compared to single phages. Phage cocktails vB_EcoMC3, vB_EcoMC4, and vB_EcoMC6 were the most efficient phages in reducing *E. coli* O177 cells (66.67%, 66.92% and 66.42%, respectively). However, the highest percentage reduction in cell counts was observed in samples treated with single phages vB_EcoM_366B and vB_EcoM_3A1 (62.31% and 62.74%, respectively). This variation in percentage cell counts reduction between the phage types could be attributed to the fact phage resistance may affect the efficacy of single phages more than phage cocktails and thus prolong the time required for the reduction of *E. coli* O177 cells [41,42,43]. In addition, phage cocktails tend to have large bust size, which may accelerate phage infections, cell lysis and complete elimination of the pathogen within a short period of time. The variation between single phages and phage cocktails’ burst sizes was observed in our previous studies [44,45]. Nonetheless, the efficacy both phage types was lower than that of the previous study by Rivas et al. [36] who reported that single phages reduced *E. coli* O157 cell counts to below the detection limit within 4 h of incubation. Different fermentation substrates and target bacteria (*E. coli* O157) could be the reasons for the difference in phage efficacy.

## 4. Materials and Methods

### 4.1. Ethical Statement

A rumen fistulated cow was used as a donor for the rumen fluid used in this study. The study was approved by the Animal Care Research Ethics Committee (AnimCare REC), North-West University (Certificate no: NWU-01223-19-S9).

### 4.2. Preparation of Grass Hay Substrate for Ex Vivo Rumen Fermentation

Milled *Eragrotis plana* hay was used as a substrate in the in vitro ruminal fermentation system. The grass hay was autoclaved in a 2 L beaker at 121 °C and 15 PSI for 15 min. After autoclaving, the grass was stored in the beaker covered with an aluminium foil at a room temperature until use. Sterile serum bottles (125 mL) were used as fermentation vessels into which 1 g of the sterile grass hay substrate and 90 mL fermentation buffer (pH 6.8) were added. The mixture was temperature acclimated at 39 °C for 18 h before inoculation with rumen fluid.

### 4.3. Bacterial Strain

Shiga toxigenic MDR atypical enteropathogenic *E. coli* O177 strain was obtained from our bacterial culture collection. Characterisation of this strain has been described in our previous study [3]. Briefly, a previously frozen (−80 °C) stock culture of MDR *E. coli* O177 strain was revived on MacConkey agar. A single colony was then transferred to a sterile falcon tube (50 mL) with nutrient broth (10 mL) and incubated for 24 h at 37 °C.

### 4.4. Propagation of Phage Lysates

Eight single phages (vB_EcoM_10C2, vB_EcoM_10C3, vB_EcoM_118B, vB_EcoM_11B2, vB_EcoM_12A1, vB_EcoM_366B, vB_EcoM_366V, and vB_EcoM_3A1) and six phage cocktails (vB_EcoMC1, vB_EcoMC2, vB_EcoMC3, vB_EcoMC4, vB_EcoMC5, and vB_EcoMC6) were used to challenge *E. coli* O177 under simulated ruminal fermentation conditions. Biological characteristics of the eight single phages used in this study have been described in our previous study [44]. Prior to conducting the ex vivo rumen simulation assay, single phage stock solution was propagated using shiga toxigenic MDR *E. coli* O177 host as previously described [46]. Phage cocktail formulation was prepared using single phages as previously described [45]. Final titration of both single phages and cocktails was determined using double-layer agar method and the working titres were adjusted to 1 × 10^8^ PFU/mL.

### 4.5. Efficacy of Phages Against E. coli O177 Strain in Ex Vivo Rumen Fermentation Conditions

#### 4.5.1. Collection of Rumen Fluid

The viability and efficacy of phages under anaerobic rumen fermentation conditions were evaluated using three runs of the Reading Pressure Technique [47]. Briefly, rumen fluid was obtained from the donor (a five-year old rumen cannulated Bonsmara cow) before morning feeding. Rumen digesta was collected into a sterile prewarmed thermos flask and then transported to the laboratory. Upon arrival, the fluid was filtered using three layers of cheesecloth. The filtrate was kept in a water bath at 39 °C with constant purging with carbon dioxide gas (CO_2_). An aliquot of 10 mL rumen fluid was added into each of the 120 mL serum bottles containing 90 mL buffer and 1 g of grass hay substrate. The bottles were purged with CO_2_ and sealed with butyl stopper rubbers.

#### 4.5.2. *E. coli* O177 and Phage Inoculation

Eighty serum bottles containing 1 g grass hay substrate, 90 mL fermentation buffer and 10 mL rumen fluid were used per run. Of these, 70 serum bottles were inoculated with 200 µL *E. coli* O177 (1 × 10^8^ CFU/mL). Subsequently, samples were treated with 2000 µL of each phage (1 × 10^8^ PFU/mL) (5 replicates per phage (40 for single phages and 30 for phage cocktails)) to achieve multiple of infection (MOI) of 10. Two sets of five control bottles containing grass hay substrate and buffered rumen fluid were designed as follows: 1. serum bottles inoculated with 200 µL *E. coli* O177 but without phage treatment and 2. serum bottles with neither *E. coli* O177 nor phage treatments. All the bottles were purged with CO_2_, sealed with butyl stopper rubbers, and incubated at 39 °C for 48 h.

### 4.6. Determination of Phage Titres and E. coli O177 Cell Counts

Samples of serum bottle contents were taken at 0, 6, 12, 24, 36, 48 h post-inoculation for enumeration of viable *E. coli* O177 cell counts and titration of single phages and phage cocktails. An aliquot of 5 mL was withdrawn from each bottle using a 20 mL syringe and transferred into 50 mL sterile falcon tubes. New syringes were used for each sampling. The samples were centrifuged at 10,000× *g* for 10 min at 4 °C to sediment the bacteria. The pellets were resuspended in 1.5 mL phosphate buffer saline (PBS, pH 7.0) and the mixture was vortexed for 2 min to obtain a homogeneous suspension. A 10-fold serial dilution of each sample was prepared. Subsequently, 100 µL from the dilutions was plated in triplicate on MacConkey agar supplemented with 1 mg/L potassium tellurite, 50 µg/mL nalidixic acid, 50 µg/mL streptomycin sulfate, and 50 µg/mL erythromycin [36]. The plates were incubated at 37 °C for 24 h and colony count was performed as previously described [48]. Average number of viable *E. coli* O177 cells recovered was calculated using the formula: [(n_1_ − n_2_) * d], where n_1_ = average number of colonies counted from negative control samples, n_2_ = average number of colonies counted from the samples treated with phages and d = dilution factor. The supernatant, which contained phages, was filter-sterilised using 0.22 µm pore-size syringe filter. Phage lysates (single phages and phage cocktails) were serially diluted using sterile PBS (pH 7.0) and their titrations were determined using double-layer agar method [49]. All the assays were performed in triplicate and phage titres were expressed as PFU/mL.

### 4.7. Statistical Analysis

Efficacy of phages was analysed [50] according to the following general linear model:Yij = µ + Ti + Eij
where Yij is the dependent variable ij; µ is the overall mean; Ti is the effect of phages and Eij is the random error associated with ij. Phage viability and phage efficacy over incubation time was analysed for linear and quadratic trends (PROC RSREG; [50]). For all statistical tests, significance was declared at *p* < 0.05 and the least square means were separated using the probability of difference option in SAS.

## 5. Conclusions

This study provides the evidence that lytic single *E. coli* O177-specific phages and their cocktails were stable and viable under simulated rumen fermentation conditions. Although both phage types significantly reduced *E. coli* O177 cell counts in the rumen model, phage cocktails were more efficacious compared to single phages. These findings suggest that there is potential for in-feed *E. coli* O177-specific phage cocktails to be offered to feedlot finished beef cattle at least two days prior to slaughter to reduce the risks of meat contamination. However, more work needs to be done to determine phage–bacterium interaction in live animals.

## Figures and Tables

**Figure 1 antibiotics-10-00265-f001:**
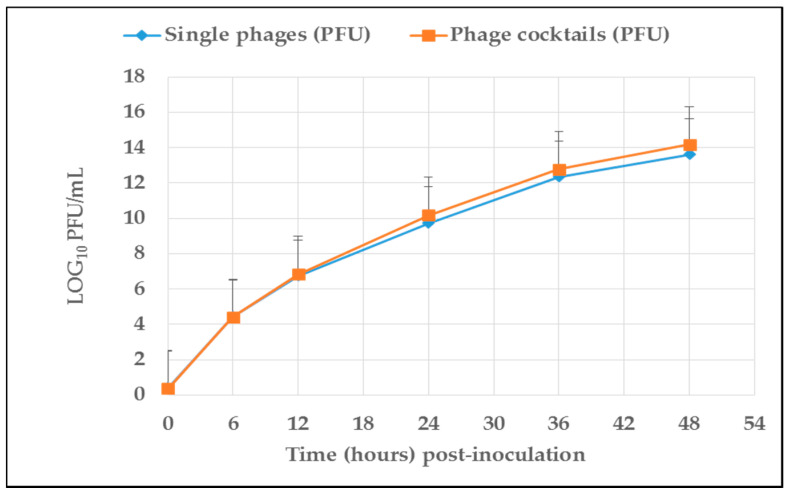
Average concentration of single phages and phage cocktails (PFU/mL) recovered from rumen contents at different incubation times. Standard deviation is represented by the error bars.

**Table 1 antibiotics-10-00265-t001:** Relationship between single phage titres (*y*) and in vitro ruminal incubation time (*x*).

Single Phage	Regression Equation	R^2^	*p*-Value	Predicted Time to Peak Titre (h)
vB_EcoM_10C2	y = 1.128 (±0.2139) + 0.501 (±0.0237)*x* − 0.005 (±0.0005)*x*^2^	0.985	<0.001	50
vB_EcoM_10C3	y = 0.752 (±0.1727) + 0.525 (±0.0191)*x* − 0.005 (±0.0004)*x*^2^	0.991	<0.001	53
vB_EcoM_118B	y = 1.149 (±0.1815) + 0.496 (±0.0201)*x* − 0.005 (±0.0004)*x*^2^	0.989	<0.001	50
vB_EcoM_11B2	y = 1.036 (±0.1403) + 0.502 (±0.0155)*x* − 0.005 (±0.0003)*x*^2^	0.994	<0.001	50
vB_EcoM_12A1	y = 0.730 (±0.1725) + 0.521 (±0.0191)*x* − 0.005 (±0.0004)*x*^2^	0.991	<0.001	52
vB_EcoM_366B	y = 1.014 (±0.2037) + 0.503 (±0.0226)*x* − 0.005 (±0.0005)*x*^2^	0.987	<0.001	50
vB_EcoM_366V	y = 1.088 (±0.2153) + 0.503 (±0.0238)*x* − 0.005 (±0.0005)*x*^2^	0.985	<0.001	50
vB_EcoM_3A1	y = 1.041 (±0.2006) + 0.489 (±0.0222)*x* − 0.005 (±0.0004)*x*^2^	0.811	<0.001	49

**Table 2 antibiotics-10-00265-t002:** Relationship between titres phage cocktails (*y*) and in vitro ruminal incubation time (*x*).

Phage Cocktail	Regression Equation	R^2^	*p*-Value	Predicted Time to Peak Titre (h)
vB_EcoMC1	y = 0.981 (±0.1845) + 0.514 (±0.0204)*x* − 0.005 (±0.0004)*x*^2^	0.990	<0.001	51
vB_EcoMC2	y = 0.682 (±0.1525) + 0.535 (±0.0169)*x* − 0.005 (±0.0003)*x*^2^	0.993	<0.001	54
vB_EcoMC3	y = 0.919 (±0.1368) + 0.532 (±0.0151)*x* − 0.005 (±0.0003)*x*^2^	0.995	<0.001	53
vB_EcoMC4	y = 0.531 (±0.1548) + 0.548 (±0.0171)*x* − 0.005 (±0.0003)*x*^2^	0.994	<0.001	55
vB_EcoMC5	y = 1.020 (±0.2435) + 0.519 (±0.0270)*x* − 0.005 (±0.0005)*x*^2^	0.982	<0.001	52
vB_EcoMC6	y = 1.145 (±0.2101) + 0.527 (±0.0233)*x* − 0.005 (±0.0005)*x*^2^	0.987	<0.001	53

**Table 3 antibiotics-10-00265-t003:** Effect (% reduction) of single phages on *E. coli* O177 cells in a rumen fermentation model.

Time (h)	Single Phages								
	vB_EcoM_10C2	vB_EcoM_10C3	vB_EcoM_118B	vB_EcoM_11B2	vB_EcoM_12A1	vB_EcoM_366B	vB_EcoM_366V	vB_EcoM_3A1	SEM
6	17.25 ^c^	16.58 ^abc^	16.31 ^ab^	16.89 ^bc^	16.03 ^a^	19.41 ^e^	18.09 ^d^	16.46 ^ab^	0.168
12	27.03 ^b^	28.59 ^d^	25.86 ^a^	27.66 ^bc^	28.05 ^cd^	28.61 ^d^	27.92 ^cd^	27.54 ^bc^	0.194
24	48.37 ^a^	48.53 ^b^	49.56 ^f^	50.10 ^g^	49.60 ^f^	49.52 ^e^	49.07 ^c^	49.50 ^d^	0.123
36	54.37 ^a^	54.43 ^a^	55.13 ^ab^	55.62 ^bc^	54.96 ^ab^	56.10 ^c^	55.78 ^bc^	56.42 ^c^	0.207
48	61.02 ^a^	61.12 ^a^	61.69 ^ab^	61.73 ^ab^	61.89 ^b^	62.31 ^bc^	61.90 ^b^	62.74 ^c^	0.180

Means with different superscripts (a, b, c, d, e, f and g) within a rows are significantly different (*p* < 0.05), SEM = standard error mean. The percentage reduction was calculated based on the difference between the mean log_10_
*E. coli* cell counts (CFU/mL) in phage-treated samples and mean log_10_
*E. coli* cell counts (CFU/mL) in untreated control samples for each phage and at each time point.

**Table 4 antibiotics-10-00265-t004:** Effect (% reduction) of phage cocktails on *E. coli* O177 cells in a rumen fermentation model.

Time (h)	Phage Cocktails						
	vB_EcoMC1	vB_EcoMC2	vB_EcoMC3	vB_EcoMC4	vB_EcoMC5	vB_EcoMC6	SEM
6	21.79 ^b^	20.65 ^a^	24.38 ^c^	26.63 ^d^	21.05 ^a^	24.65 ^c^	0.168
12	32.52 ^b^	30.29 ^a^	34.83 ^c^	36.75 ^d^	30.73 ^a^	34.69 ^c^	0.194
24	50.81 ^b^	51.51 ^c^	54.34 ^e^	53.51 ^d^	50.27 ^a^	55.19 ^f^	0.123
36	58.60 ^b^	58.67 ^b^	59.84 ^c^	59.81 ^c^	56.41 ^a^	60.69 ^d^	0.207
48	65.34 ^b^	62.82 ^a^	66.67 ^c^	66.92 ^c^	62.35 ^a^	66.42 ^c^	0.180

Means with different superscripts (a, b, c, d, e and f) within a rows are significantly different (*p* < 0.05), SEM = standard error mean. The percentage reduction was calculated based on the difference between the mean log_10_
*E. coli* cell counts (CFU/mL) in phage cocktail-treated samples and mean log_10_
*E. coli* cell counts (CFU/mL) in untreated control samples for each phage cocktail and at each time point.

**Table 5 antibiotics-10-00265-t005:** Relationship between in vitro ruminal incubation time (*x*) and percent reduction in *E. coli* O177 cells (*y*) when challenged with single phages.

Single Phages	Regression Equation	R^2^	*p*-Value	Time to Maximum Effect (h)
vB_EcoM_10C2	y = 1.212 (±0.6744) + 2.548 (±0.0746)*x* − 0.027 (±0.0015)*x*^2^	0.993	<0.001	48
vB_EcoM_10C3	y = 1.258 (±0.650) + 2.583 (±0.0720)*x* − 0.028 (±0.0014)*x*^2^	0.994	<0.001	47
vB_EcoM_118B	y = 0.476 (±0.7273) + 2.599 (±0.0805)*x* − 0.027 (±0.0016)*x*^2^	0.993	<0.001	48
vB_EcoM_11B2	y = 1.054 (±0.6771) + 2.646 (±0.0749)*x* − 0.029 (±0.0015)*x*^2^	0.993	<0.001	47
vB_EcoM_12A1	y = 0.715 (±0.677) + 2.636 (±0.0749)*x* − 0.028 (±0.0015)*x*^2^	0.993	<0.001	48
vB_EcoM_366B	y = 1.930 (±0.7607) + 2.621 (±0.084)*x* − 0.028 (±0.0017)*x*^2^	0.992	<0.001	47
vB_EcoM_366V	y = 1.317 (±0.6620) + 2.616 (±0.0732)*x* − 0.028 (±0.0014)*x*^2^	0.994	<0.001	47
vB_EcoM_3A1	y = 0.680 (±0.5571) + 2.636 (±0.0616)*x* − 0.028 (±0.0012)*x*^2^	0.996	<0.001	47

**Table 6 antibiotics-10-00265-t006:** Relationship between in vitro ruminal incubation time (*x*) and percent reduction in *E. coli* O177 cells (*y*) when challenged with phage cocktails.

Phage Cocktails	Regression Equation	R^2^	*p*-Value	Time to Maximum Effect (h)
vB_EcoMC1	y = 3.359 (±0.938) + 2.697 (±0.1038)*x* − 0.029 (±0.0021)*x*^2^	0.988	<0.001	46
vB_EcoMC2	y = 1.887 (±0.734) + 2.815 (±0.0813)*x* − 0.032 (±0.0016)*x*^2^	0.993	<0.001	44
vB_EcoMC3	y = 3.803 (±1.2326) + 2.905 (±0.1364)*x* − 0.034 (±0.0027)*x*^2^	0.981	<0.001	43
vB_EcoMC4	y = 5.049 (±1.510) + 2.864 (±0.167)*x* − 0.033 (±0.0033)*x*^2^	0.970	<0.001	44
vB_EcoMC5	y = 2.671 (±0.899) + 2.692 (±0.0995)*x* − 0.030 (±0.0020)*x*^2^	0.988	<0.001	45
vB_EcoMC6	y = 3.606 (±1.1709) + 2.977 (±0.1296)*x* − 0.035 (±0.0026)*x*^2^	0.983	<0.001	43

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
