# Peer review of "An Assessment of the Viability of Lytic Phages and Their Potency against Multidrug Resistant Escherichia coli O177 Strains under Simulated Rumen Fermentation Conditions"

_antibiotics, 2021, doi:10.3390/antibiotics10030265_

Round 1

Reviewer 1 Report

The authors presented one very interesting and innovative study of the viability of lytic phages and their potency against multidrug resistant Escherichia coli O177 strains under simulated rumen fermentation conditions. Overall the manuscript was well elaborated and well written, and the results were clearly presented showing the possible efficacy of the phages to control multidrug resistant E.coli O177 strains growth, showing expressive reduction either with isolate phages, or combination of phages in a model that tries to mimic rumen conditions.  

I think there is one question that the authors should answer to improve the manuscript.

  • Tables 3 and 4 shows the results of inhibition of growth in the presence of either single phages, or combination of phages. Despite the tables presented different treatments, and different values, surprisingly  the standard deviation error of the tables are exact the same. I’m not sure if I understand correctly these tables, or if there is some problem with these results.   

Author Response

Response to Reviewer 1 Comments

Point 1: Tables 3 and 4 shows the results of inhibition of growth in the presence of either single phages, or combination of phages. Despite the tables presented different treatments, and different values, surprisingly the standard deviation error of the tables are exact the same. I’m not sure if I understand correctly these tables, or if there is some problem with these results.

Response 1: In Tables 3 and 4, we report the estimated standard error of the mean (SEM), calculated by dividing the standard deviation by the square root of the sample size for each incubation period. In addition, the treatments (phages) were equally replicated, this explains why the SEM is the same for treatments within an incubation time point.

Reviewer 2 Report

Montso and colleagues assessed the use of phages against a multidrug resistant E. coli O177 strain under simulated rumen fermentation conditions. While studies as this one are important to evaluate the possibilities of using phages to prevent foodborne contamination in different conditions, this work is too simplistic and could easily be improved with some additional experiments. I therefore recommend publication only after some major changes, as detailed below.

Major comments

  1. The regression equations are in my opinion useless here. They do not add any additional information to the data, as they are simply equations fit to data points, and not modelling of the infection. I would therefore advise the authors to remove these (Tables 1, 2, 5 and 6).
  2. The authors claim to have tested viability of the phages in simulated rumen fermentation conditions but I do not see this in their results. Viability of the phages in a particular culture condition needs to be accessed in the absence of bacteria – the authors should determine phage viability in rumen fermentation conditions compared to normal culture conditions and see if phages and phage cocktails remain stable for long periods of time.
  3. Figure 1 and Figure 2, with the exception of phage concentration) show the same data as tables 3 and 4 – the effect of phages in bacterial concentration overtime. I would recommend that the authors put this (Figures 1 and 2, and Tables 3 and 4) in one single Figure (not table) with two panels – one panel for reduction achieved with individual phages and the second panel for reduction achieved with phage cocktails.
  4. As it is, the manuscript simply shows that phages can kill bacteria in simulated rumen fermentation conditions. I believe it would make the paper much stronger if the authors would address the following questions: Does resistance to the phages develop during treatment? 2. When does resistance start to emerge? Is this different for individual phages versus phage cocktails?
  5. In line 187, the authors mention that phage cocktails tend to have large burst size than single phages. Have they tested this for their single phages and phage cocktails?

Minor comments

Line 27 – the authors mention 73.25% as the best reduction achieved, but lines 29-30 do not mention which phage or phage cocktail resulted in this reduction.

Line 39 – stx should be explained.

Line 56 – first mention of phages, so it should be written as bacteriophages.

Line 58 – change to “Phages are viruses that infect”

Line 66 – change to “the use of phages against shiga-toxigenic”

Line 68 – Why are strains O177 relevant? The authors should explain this.

Line 78 – “rumen simulation assay after.” This sentence is missing something.

Figure 1 and 2 – while I recommend in my major comments above that figures should be changed, I would like to point out for future reference that the symbols and lines referring to the same condition should have the same colour. Also, it was unclear which phage and phage cocktail was used to generate this data. The Legend of Figure 1 is also wrong, because it mentions the use of phage cocktails while the figure refers to single phages.

Line 162 – the authors cannot compare their results to that of Bach because the incubation period was very different. They can only compare if they compare to the same time points of that study.  

Author Response

Response to Reviewer 2 Comments

Point 1: The regression equations are in my opinion useless here. They do not add any additional information to the data, as they are simply equations fit to data points, and not modelling of the infection. I would therefore advise the authors to remove these (Tables 1, 2, 5 and 6).

Response 1: Yes, the regression equations are not modelling infection (we hope we did not inadvertently suggest otherwise in the paper), we merely present them as statistical tools to describe/define the underlying relationships (trends or patterns) in the observed range of phage titres and E. coli O177 cells in response to incubation time. The equations allow us to derive the different times at which phages and phage cocktails attained peak titres and the amount of time that phages need to exhibit maximum efficacy against E. coli in the simulated rumen. Identifying these time points is vital because they tell us whether the phages will have sufficient time to attain peak titres and thus effectively reduce E. coli loads in the rumen of a live animal, given that rumen retention times vary with the animal’s diet. Therefore, we would like to keep the equations because presenting these predicted incubation time points without the equations from which they are derived would not be helpful to the reader.

Point 2: The authors claim to have tested viability of the phages in simulated rumen fermentation conditions but I do not see this in their results. Viability of the phages in a particular culture condition needs to be accessed in the absence of bacteria – the authors should determine phage viability in rumen fermentation conditions compared to normal culture conditions and see if phages and phage cocktails remain stable for long periods of time.

Response 2: The stability and viability of these phages in the absence of bacteria have been tested under various culturing conditions and published in our previous study “Montso, P.K.; Mlambo, V.; Ateba, C.N. 2019. Characterisation of lytic bacteriophages infecting multi-drug resistant shiga-toxigenic atypical Escherichia coli O177 strains isolated from cattle faeces. Front. Public Health. 7:355”. However, we believe that even results from the current study confirm the viability of phages in simulated rumen fermentation conditions. Indeed, revised Figure 1 shows increasing phage titers (PFU/mL) over time in the simulated rumen fermentation system – if the phages were not viable we would expect to observe no change in PFUs over time or decline in PFUs over time — depending on detection or assay method used.

Point 3: Figure 1 and Figure 2, with the exception of phage concentration) show the same data as tables 3 and 4 – the effect of phages in bacterial concentration overtime. I would recommend that the authors put this (Figures 1 and 2, and Tables 3 and 4) in one single Figure (not table) with two panels – one panel for reduction achieved with individual phages and the second panel for reduction achieved with phage cocktails.

 Response 3: We agree with your observation that Figures 1 and 2 present a summary of the data presented in Tables 3 and 4 so we have resolved that by removing phage efficacy data from the two Figures to produce just one Figure that report phage concentrations over tie.  An attempt was made to convert Table 3 and 4 into one single Figure (line graph) with two panels as suggested by the reviewer. However, the new Figure became too cluttered and hard to read — with some lines (for the treatments) stacked on top of each other. We have, therefore, elected to keep the Tables (3 and 4).

Point 4: As it is, the manuscript simply shows that phages can kill bacteria in simulated rumen fermentation conditions. I believe it would make the paper much stronger if the authors would address the following questions: Does resistance to the phages develop during treatment? 2. When does resistance start to emerge? Is this different for individual phages versus phage cocktails?

Response 4: We agree, additional data on phage resistance would be useful but phage resistance was not assessed during this study as we needed some empirical evidence that the phages would be viable under the described conditions, in the first instance. We think that possible phage resistance under simulated rumen fermentation conditions is the next logical research question for us.

Point 5: In line 187, the authors mention that phage cocktails tend to have large burst size than single phages. Have they tested this for their single phages and phage cocktails?

Response 5: Indeed, the burst size of the single phages has tested in our previous study “Montso, P.K.; Mlambo, V.; Ateba, C.N. 2019. Characterisation of lytic bacteriophages infecting multi-drug resistant shiga-toxigenic atypical Escherichia coli O177 strains isolated from cattle faeces. Front. Public Health. 7:355”. Although the data was not included in this paper (a reference has been provided), the burst size of phage cocktails have been determined prior to the experiment conducted in this study. This was performed to ensure equal concentrations of single phages and phage cocktails.

 Point 6: Line 27 – the authors mention 73.25% as the best reduction achieved, but lines 29-30 do not mention which phage or phage cocktail resulted in this reduction.

Response 6: This has been corrected in the revised paper and the new sentence reads: Phage-induced reduction of E. coli O177 cell counts reached maximum values of 61.02 - 62.74% and 62.35 – 66.92% for single phages and phage cocktails, respectively (line 27-29).

 Point 7: Line 39 – stx should be explained.

Response 7: The stx has now been described in the paper and the new sentence now reads: ‘…. harbour stx operon (stx1 and stx2) responsible for production of shiga toxin’.

 Point 8: Line 56 – first mention of phages, so it should be written as bacteriophages.

Response 8: ‘Phage’ has been replaced with ‘bacteriophages’

 Point 9: Line 58 – change to “Phages are viruses that infect”

 Response 9: This has been changed as suggested and new sentence now reads: ‘Phages are viruses that infect and kill bacteria cells through a lytic process [19,20] (Line 60).

Point 10: Line 66 – change to “the use of phages against shiga-toxigenic”

 Response 10: This has been changed as suggested and the new sentence now reads: “Furthermore, studies with live animals have focused on the use of phages against shiga-toxigenic E. coli O157 and other non O157 serotypes [26] (Line 67-68).

Point 11: Line 68 – Why are strains O177 relevant? The authors should explain this.

Response 11: The relevance of this strain has now been explained in the paper (Line 69-79) as follows: “The selection of this strain was motivated by the fact that it harbours virulence and antibiotic resistance genes (3). Furthermore, its presence in cattle may result into food contamination, especially during slaughter.”

Point 12: Line 78 – “rumen simulation assay after.” This sentence is missing something.

Response 12: The sentence has been revised and it now reads: “Figure 1 presents the number (mean log10) of phages (PFU/mL) recovered from artificial rumen contents over a 48-hour incubation period. (Line 80-81).”

 Point 13: Figure 1 and 2 – while I recommend in my major comments above that figures should be changed, I would like to point out for future reference that the symbols and lines referring to the same condition should have the same colour. Also, it was unclear which phage and phage cocktail was used to generate this data. The Legend of Figure 1 is also wrong, because it mentions the use of phage cocktails while the figure refers to single phages.

Response 13: The colour of the symbols and lines have been changed such that the symbols and lines referring to the same condition have the same colour. The phages (single phages and phage cocktails) used to generate the data have been listed in Table 7. The Legend of the figures has been corrected accordingly.

 Point 14: Line 162 – the authors cannot compare their results to that of Bach because the incubation period was very different. They can only compare if they compare to the same time points of that study.

Response 14: As suggested, we have now revised this statement to compare phage titres at the same incubation time (48 h) across the two studies.

Round 2

Reviewer 2 Report

The authors have answered all my previous comments and made the necessary changes to the manuscript. The only point that remains is that the exact single phage and phage cocktail that are represented in Figure 1 should be indicated in the Legend. Authors mention that this is indicated in Table 7, but Table 7 simply lists all single phages and cocktails used in this work. From what I understand from Figure 1, only one phage and one cocktail are represented. It should be made clear which phage/cocktail were used. Other than this small detail, I believe the manuscript can be published.

Author Response

Point 1: The authors have answered all my previous comments and made the necessary changes to the manuscript. The only point that remains is that the exact single phage and phage cocktail that are represented in Figure 1 should be indicated in the Legend. Authors mention that this is indicated in Table 7, but Table 7 simply lists all single phages and cocktails used in this work. From what I understand from Figure 1, only one phage and one cocktail are represented. It should be made clear which phage/cocktail were used. Other than this small detail, I believe the manuscript can be published.——From Reviewer 2, please revise accordingly.

Response 1: The Figure shows the average of all single phages and phage cocktails (PFU/mL) at each time point over a period of 48 hours. Therefore, the Figure legend have been revised and reads: Average single phages and phages cocktails (PFU/mL) recovered from rumen contents, Standard deviation is represented by the error bars.